# High resolution clonal architecture of hypomutated Wilms tumours

Henry Lee-Six [1,2,3,17], Taryn D. Treger[1,3,4,17], Manas Dave[1,5,6], Tim HH Coorens [7], Nathaniel D. Anderson[1], Yvonne Tiersma[8,9], Sepide Derakhshan[8,9], Sanne de Haan[8,9], Marry M. van den Heuvel-Eibrink [8], Yichen Wang [1], Anna Wenger [1], Reem Al-Saadi [10,11], Alice Lawford[11], Aleksandra Letunovska[10,11], Jenny Wegert [12], Conor Parks [1], Guillaume Morcrette[10,11], Manfred Gessler [12], Gordan Vujanic [13], Tanzina Chowdhury[10,11], Maureen J O'Sullivan[14,15], Ronald R. de Krijger[8,16], Michael R. Stratton [1], Kathy Pritchard-Jones [10], J. Ciaran Hutchinson [11] ✉, Jarno Drost [8,9] ✉ & Sam Behjati [1,3,4] ✉

A paradigm of childhood cancers is that they have a low mutation burden, with some ostensibly bearing fewer mutations than the normal tissues from which they derive. We set out to resolve this paradox by examining paediatric renal cancers with exceptionally few mutations using high resolution, high depth sequencing approaches. We find that apparent hypomutation is the result of unusual clonal architecture due to a normal tissue-like mode of tumour evolution, raising the possibility that the mutation burden of some cancers has been systematically misjudged.

The mutation burden of cancers both reflects fundamental features of tumour biology and can be of clinical significance. A high mutation burden is correlated with the number of neoantigens and has been associated with the success of immunotherapy[1], the likelihood of resistance to treatment (especially mutation-guided treatment)[2,3], and the progression to more advanced disease[4]. Studies of childhood cancers have established that they generally have few mutations[5,6]. Intriguingly, the investigation of normal tissues in recent years[7] has revealed that some age-matched normal cells have more mutations than certain paediatric malignancies, even though cancers derive from normal cells and the transition to malignancy is associated with, if anything, an increase in mutation rate[8–10]. We set out to investigate this apparent paradox, focusing our efforts on characterising cases of Wilms tumour, a childhood kidney cancer with one of the lowest mutation burdens[6].

In this work, we determine the true per-cell mutation burden of normal kidneys and Wilms tumours. We show that standard sequencing methods significantly underestimate the mutation burden of tumours of newborns, and that this is a result of their unusual normal tissue-like clonal architecture. We delineate the driver mutations that underpin such deeply branching phylogenies.

## Results

### Mutation burdens of normal kidney and kidney tumours

Wilms tumours from two published cohorts[6,11] displayed a broad range of mutation burdens (Fig. 1a). The least mutated cancers tended to

[1]Wellcome Sanger Institute, Hinxton, UK. [2]Department of Pathology, University of Cambridge, Cambridge, UK. [3]Cambridge University Hospitals NHS Foundation Trust, Cambridge, UK. [4]Department of Paediatrics, University of Cambridge, Cambridge, UK. [5]Department of Biochemistry, The University of Cambridge, Cambridge, UK. [6]Faculty of Biology, Medicine and Health, The University of Manchester, Manchester, UK. [7]Broad Institute of MIT and Harvard, Cambridge, MA, USA. [8]Princess Máxima Center for Pediatric Oncology, Utrecht, The Netherlands. [9]Oncode Institute, Utrecht, The Netherlands. [10]UCL Great Ormond Street Institute of Child Health, London, UK. [11]Great Ormond Street Hospital for Children, London, UK. [12]Theodor-Boveri-Institute/Biocenter, Developmental Biochemistry, Würzburg University & Comprehensive Cancer Center Mainfranken, Würzburg, Germany. [13]Department of Pathology, Sidra Medicine, Doha, Qatar. [14]Department of Pathology, Children's Health Ireland at Crumlin, Dublin, Ireland. [15]Histopathology Department, The University of Dublin, Trinity College, Dublin, Ireland. [16]Department of Pathology, University Medical Center Utrecht, Utrecht, The Netherlands. [17]These authors contributed equally: Henry Lee-Six, Taryn D. Treger. ✉e-mail: Ciaran.hutchinson@gosh.nhs.uk; j.drost@prinsesmaximacentrum.nl; sb31@sanger.ac.uk

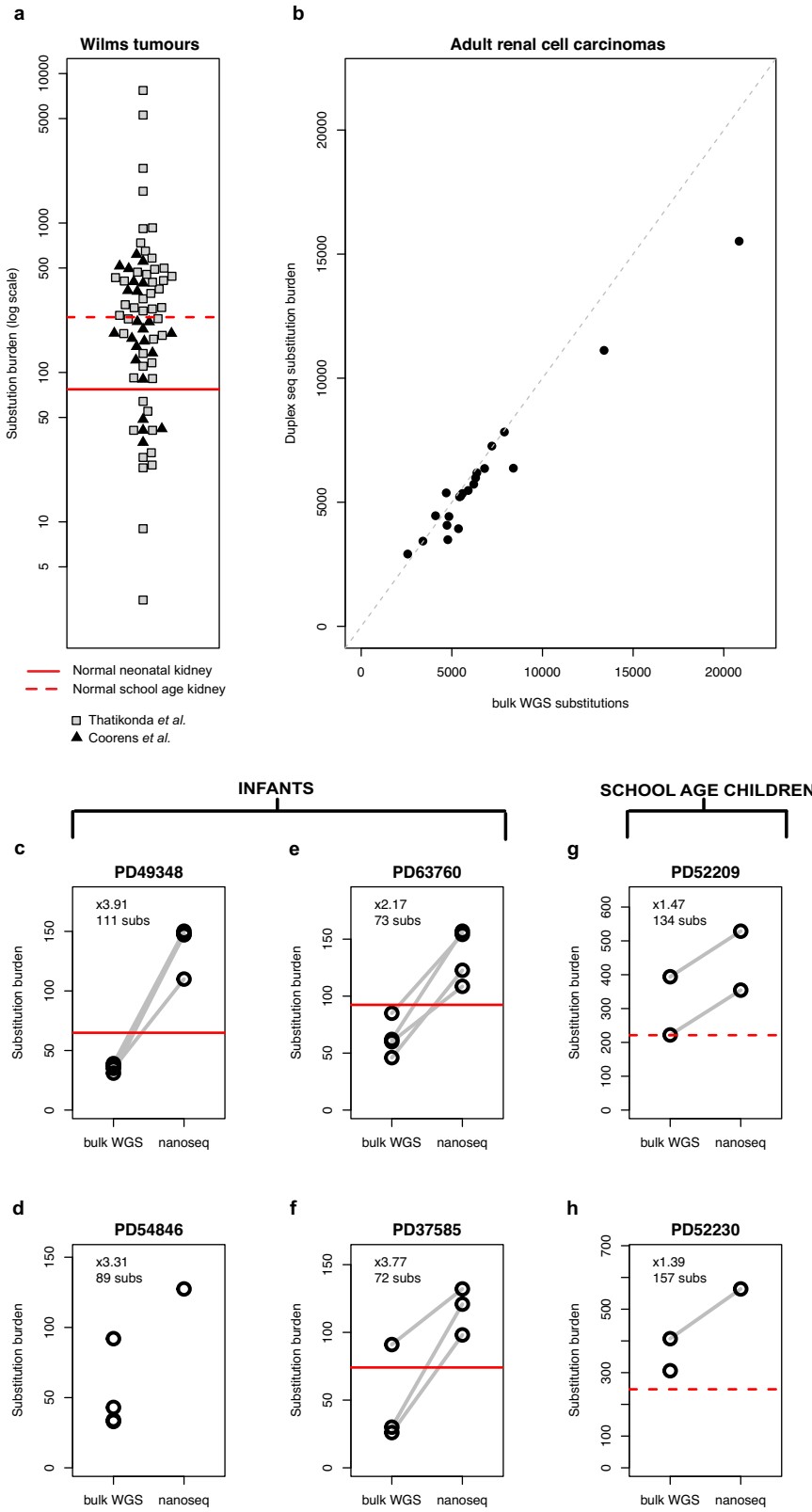

occur in the youngest children, and their low burdens could not necessarily be accounted for by technical factors (such as sequence coverage or tumour purity; Supplementary Fig. 1). As a point of reference, we sought to determine the mutation burden of normal kidneys, a value that has remained elusive due to the tissue's polyclonal composition but is now accessible through duplex sequencing. This error-corrected method allows mutations that are present on a single molecule of DNA, and therefore in a single cell, to be called accurately[12]. We identified four of the youngest children who had had resections for Wilms tumours (two newborns (PD49348, PD54846), a 4-month-old (PD63760), and a 6-month-old infant (PD37585)[13] (Methods), as well as two children diagnosed in the typical school-age range (PD52209 and PD52230). The 4-month-old and the school-age children had received neoadjuvant chemotherapy with the non-mutagenic

**Fig. 1 | Mutation burden of Wilms tumours. a** The total number of substitutions called from whole genome sequencing (WGS) of Wilms tumours from two previously published cohorts[6,11] is shown (log scale). Where there is more than one sample per tumour, the median burden across samples is displayed. A horizontal solid red line represents the mutation burden of normal neonatal kidney established by duplex sequencing (nanoseq[12]) in this study, (mean of one sample from each of three children), while the dashed red line shows that for normal kidneys of school-age children with Wilms tumours (mean of one sample from each of two children). **b** For a cohort of 21 adult renal cell carcinomas, the substitution burden derived from nanoseq is plotted against that derived by bulk WGS. **c–h** Comparison of substitution burden per diploid genome (log scale) of bulk whole genome sequencing and nanoseq for each patient's tumour. The bulk estimate has been corrected to 30× coverage, except for PD37585 (**f**) for which this was not possible (Methods, Supplementary Fig. 10). Each point represents one biopsy from a tumour, and, where the same biopsy has been sequenced using both methodologies, points are linked by grey lines. The fold change between the median burden from nanoseq and the median burden from bulk whole genome sequencing along with the absolute difference in substitutions (subs) is shown within each plot. For each case, a solid (infants) or dashed (children) red horizontal line represents a point estimate of the mutation burden derived from duplex sequencing of the normal kidney of the same patient as the tumour sample. **c** PD49348, a neonatal Wilms tumour (four biopsies studied by both bulk sequencing and nanoseq); **d** PD54846, a neonatal Wilms tumour (four biopsies studied by bulk sequencing alone and one biopsy studied by nanoseq alone); **e** PD63760, a Wilms tumour from a 4-month-old (four biopsies studied by both bulk sequencing and nanoseq); **f** PD37585, a Wilms tumour from a 6-month-old (three biopsies studied by both bulk sequencing and nanoseq); **g** PD52209, a Wilms tumour from a school-age child (two biopsies studied by both bulk sequencing and nanoseq); and **h** PD52230 a Wilms tumour from a school-age child (one biopsy studied by both bulk sequencing and nanoseq, and one biopsy studied by nanoseq alone). Source data are provided as a Source Data file.

agents vincristine and actinomycin D. For all but one (PD54846) of the six children, resected normal kidney ipsilateral to the tumour was available for duplex sequencing. Three normal neonatal kidneys had a mutation burden of 65 (PD49348), 74 (PD37585), and 92 (PD63760) mutations per diploid genome, while the normal kidneys from the two school-age children bore 221 (PD52209) and 248 (PD52230) mutations. Normal kidneys from school-age children had more mutations than half of previously published Wilms tumour genomes, and even neonatal kidney cells had more mutations than one in five Wilms tumours.

Reasoning that the lower burden of certain cancers may be the result of technical factors, we applied both bulk whole genome sequencing and duplex sequencing to a cohort of adult renal cell carcinomas[14]. A strong correlation was noted between the mutation burden derived through the two methods (Fig. 1b). This suggests that the excess number of subclonal mutations in these tumours called by duplex sequencing is relatively small compared to the number of clonal mutations called by either method, and may be offset by the ability of whole genome sequencing to call mutations that exist in parallel subclones.

For an intra-individual comparison of tumour and normal mutation burdens, duplex sequencing was applied to the Wilms tumours whose matched normal kidneys had already been assayed. As expected, given their age, after correcting for coverage (Methods), bulk whole genome sequencing of the four infant tumours revealed a low mutation burden of 36, 38.5, 61, and 30 substitutions, respectively[13](median across tumour samples). The same bulk tissue samples, re-interrogated by duplex sequencing, had a per cell burden that was up to four-fold higher, with 72–111 additional mutations called per tumour biopsy (Fig. 1c–f)[13]. This is likely a slight underestimate, as a small proportion of infiltrating immune cells may lower the mean number of mutations (Methods, Supplementary Figs. 2 and 3). Interestingly, the duplex sequencing-derived mutation burden of the normal kidney fell between the matched tumour's estimates derived from bulk whole genome and duplex sequencing (Fig. 1b, d, e), indicating that tumour cells had a slightly increased mutation rate relative to their normal counterparts.

Next, we investigated the Wilms tumours of the two school-age children with the same method. While the absolute difference in mutations detected by bulk and duplex sequencing for these tumours was similar to that of the infants, the initial bulk mutation burden of hundreds of mutations resulted in what appeared to be a minimal gain in mutation burden by duplex sequencing (Fig. 1g, h). It seems that while Wilms tumours with a low mutation burden, as ascertained by bulk sequencing, have a much higher burden by single molecule resolution sequencing, the difference is less pronounced for more mutated Wilms tumours or adult renal cancers.

## Clonal architecture of hypomutated tumours

The discrepancy between bulk and single cell resolution mutation burdens in our infant cases may be the result of a tissue's clonal architecture (Fig. 2a). Conventional bulk sequencing detects only mutations that are shared by a large proportion of cells in the sample and misses private mutations, however numerous they might be. To test whether clonal architecture was obfuscating mutation burden, we reconstructed the phylogenies of tumours with high and low mutation burdens. For two cases in which fresh tissue was available (PD37585, PD49348), organoids could be derived from single cancer cells and adjacent normal kidney tissue[15,16] (Fig. 2b, c and Supplementary Fig. 4). This allows the construction of a single cell resolution phylogeny[17]. For other cases in which sufficient frozen tissue remained (PD54846, PD52209, PD52230), each tree was built using 20–30 biopsies of ~100 cells, isolated by laser capture microdissection[13,18](Fig. 2d–f and Supplementary Fig. 5). Here, mutations that are subclonal within a microbiopsy may go undetected, but the trunk of the tumour and its main ramifications are readily apparent.

From these 101 genomes, we determined the clonal architecture of each tumour, using established analytical approaches[10,13,17,19](Methods). Hypomutated tumours were characterised by an extremely short trunk of 9–26 shared mutations, followed by private branches hundreds of mutations long (Fig. 2b–d). Their low burden from bulk sequencing, therefore, was a result of their clonal architecture, as we had posited. More highly mutated Wilms tumours had far longer trunks and large subclones (Fig. 2e, f), which can be detected by bulk sequencing.

For comparison, we re-analysed three previously published[17] adult colorectal cancers that had been investigated using both bulk sequencing and single-cell-derived organoids, and we generated new duplex sequencing data from the bulk tissues. These cases had similar mutation burdens through bulk and duplex sequencing[13](Supplementary Fig. 6), as we would expect given their long tumour trunks and large subclones (Fig. 2g).

Differing models of adult cancer evolution have been proposed, including successive rounds of clonal diversification and selection or 'big bang' theories, resulting in a range of possible phylogenetic configurations[20–22]. A cancer that evolves following a classical model of iterative driver mutations and clonal sweeps throughout its life[23,24] is expected to have a long trunk of clonal mutations, as well as some large subclones, one of which may, if sampled later, have completed a clonal sweep and become truncal (Supplementary Fig. 7). This is the pattern that we observed in the higher burden Wilms tumours (PD52209, PD52230), and which has been previously noted in the adult colorectal cancer cases (PD21928, PD23549, PD26636). In stark contrast, the short trunks of the hypomutated tumours (PD49348,

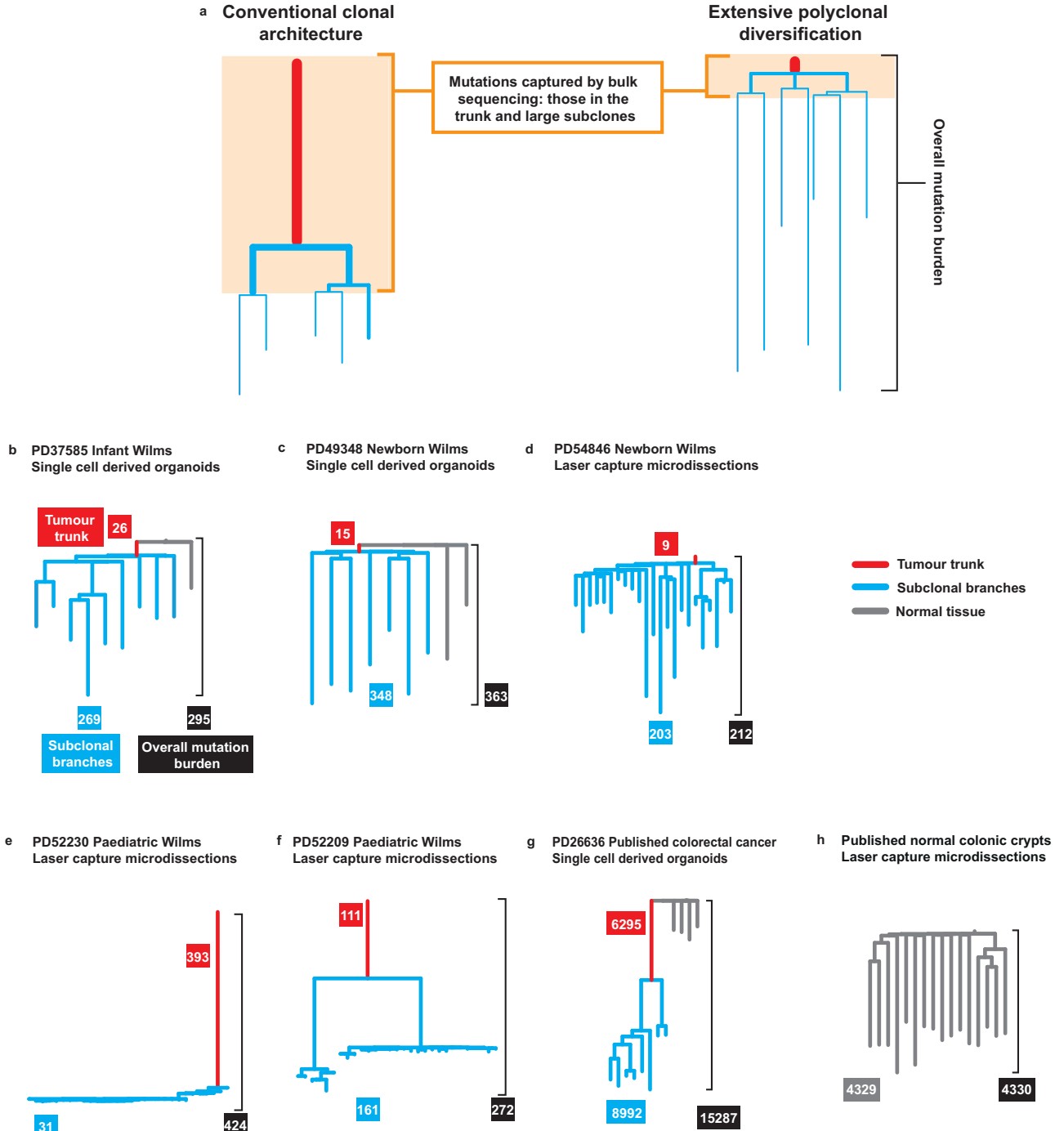

**Fig. 2 | Clonal architecture of Wilms tumours. a** A schematic illustrates that with a conventional clonal architecture (left-hand phylogeny), a high proportion of the per-tumour-cell mutation burden may be captured by bulk sequencing. In contrast, with extensive polyclonal diversification (right-hand phylogeny) only a small proportion of the per-cell mutation burden is captured. The colour key applies to all the phylogenies shown below. **b**–**f** Phylogenies of Wilms tumours, newly generated using either single-cell-derived organoids laser capture microdissections. Every tip represents either a single-cell-derived organoid or a microbiopsy. **g** Previously published[17] phylogeny of an adult colorectal cancer, constructed using whole genome sequencing of single-cell-derived organoids. **h** Previously published[10] phylogeny of normal colonic crypts from an adult, constructed using whole genome sequencing of crypts isolated by laser capture microdissection. Source data are provided as a Source Data file.

PD54846, PD37585) indicate the emergence of their most recent common ancestors in the first few months of gestation[7]. While the lack of further branching does not preclude small internecine struggles, there is no evidence for large-scale ongoing clonal competition. Indeed, the shape of the phylogeny is more reminiscent of certain normal tissues than a cancer[10,25] (Fig. 2h).

## Driver mutations of infant Wilms tumours

Analysis of the driver mutations in each tumour corroborates the phylogenetic evidence. The heavily mutated Wilms tumours and adult colorectal cancers have more than one clonal and some subclonal *bonafide* driver mutations (Supplementary Fig. 8). In contrast, in all four hypomutated tumours (PD49348, PD54846, PD63760,

PD37585) a single clonal driver was detected and no significant subclonal drivers could be identified, despite an exhaustive search of DNA and RNA sequences and DNA methylation (Methods). The oldest of these (PD37585) bore an embryonic mosaic mutation in one copy of the *REST* tumour suppressor gene, which could be detected at a low allele fraction in other normal tissues (Supplementary Fig. 9). The second copy of the gene was inactivated in the tumour trunk. On this mosaic background, therefore, only a single hit was necessary to generate the cancer. Remarkably, the tumours of the other three children (PD49348, PD54846, and PD63760), all of whom presented with Wilms tumours in the first few months of life, bore a single discernible driver mutation: a rearrangement of *FOXR2*. *FOXR2* is an emerging oncogene in certain childhood cancers[26–28] but has not previously been implicated in Wilms tumour. Akin to its structural variants in other childhood cancers, the rearrangements juxtaposed *FOXR2* to a new promoter (Fig. 3a–c) and led to its overexpression as well as increased MYCN protein, which FOXR2 stabilises (Fig. 3d)[27]. Searching for further tumours with *FOXR2* rearrangement by screening RNA sequencing (RNAseq) data of 264 Wilms tumours for *FOXR2* expression, we identified an additional tumour in an older child (Fig. 3e). The tumour exhibited additional driver events (including *TP53* p.R337C with loss of heterozygosity) and a mutation burden commensurate with age. Remarkably, the three infant tumours that shared *FOXR2* rearrangement, but not the older *FOXR2*-mutant case, were also characterised by an unusual fibro-adenomatoid morphological pattern (Fig. 3f–h, Supplementary Note). *FOXR2*-mutant infant cases clustered together and away from the older *FOXR2*-mutant case by bulk RNAseq analysis (Fig. 3i). It is possible, therefore, that *FOXR2* rearrangement delineates a particular variant of Wilms tumour in the infant context.

Our infant tumours, thus, had both an exceptionally low mutation burden and exceptionally few driver mutations. To explore whether these features were related, we expanded our driver annotation to a broader cohort of Wilms tumours of different ages. We noted a correlation between patient age, number of driver mutations, and mutation burden[13] (Fig. 3j). We suggest the following intuitive model to explain this correlation. If, as it appears to be the case in the four infant cases, a single particular driver mutation in an early embryonic cell suffices to seed a tumour, with no need for further selective sweeps in a growing organ, a tumour with a comb-like phylogeny will emerge. The short trunk will result in a low mutation burden from bulk sequencing. If, however, multiple driver mutations are required to transform an older cell, each engendering a clonal sweep and so lengthening the tumour trunk, the bulk-derived burden will be inflated (Supplementary Fig. 7).

## Discussion

In this study, we have asked the question of why certain Wilms tumours have surprisingly few mutations. Our results show that this is an artefact of standard sequencing methods. Considering the millions of cells within a tumour, each of which may have >100 mutations that are private or shared with only a small subset of other cells, the total number of mutations across all the cells in these tumours will be in the hundreds of millions (Supplementary Note). As shown by our comparison with older children and adult colorectal cancers, the similarity of bulk estimates to the true per cell burden depends on the tumour's clonal architecture. In previous studies, Wilms tumours, as examined primarily by copy number analysis[29–31], have shown variable patterns of intra-tumour heterogeneity, and so their range in mutation burden may, in part, reflect this variability. In our study, the size of which was limited by the depth of our investigations, cancers of the youngest children, with the fewest driver mutations and the most deeply branching phylogenies, are most vulnerable to an underestimate. In a future in which treatment decisions may be guided by the mutation burden and clonal structure of tumours[32,33], our findings indicate that

conventional approaches may not give an accurate picture of mutation burden, especially for cancers that follow unusual evolutionary trajectories.

## Methods

### Samples

Samples were obtained from either the Princess Máxima Center biobank in the Netherlands, or from the SIOP2001 (EudraCT clinical trial number 2007-004591-39) or IMPORT, now known as UMBRELLA, studies. The Máxima biobank protocol was approved by the Medical Ethics Committee of the Erasmus Medical Center in Rotterdam, The Netherlands, under reference number MEC-2016-739. Approval for use of the subject's data within the context of this study was granted by the Máxima biobank and data access committee (https://research.prinsesmaximacentrum.nl/en/core-facilities/biobank-clinical-research-committee), biobank request number PMCLAB2018-006. The IMPORT study was approved by the NHS National Research Ethics Service Committee London–London Bridge (12/LO/0101). The SIOP2001 study was approved by East Midlands–Derby Research Ethics Committee (National Research Ethics Service (NRES) in the UK) (reference approval number MREC/01/4/086, from 17 January 2002). Written or signed informed consent to participate, including use of tissue samples for research, was obtained from all the participants in both studies or their legal guardians. Participants were not compensated.

Patients PD49348, PD37585, and PD54846 did not receive pre-operative chemotherapy. PD63670, PD52209 and PD52230 had received pre-operative chemotherapy with vincristine and actinomycin D, which are thought to be non-mutagenic. The tumours of PD52209 and PD52230 showed little response in tumour volume, while the tumour of PD63760 shrank by ~40%.

### Statistics and reproducibility

No statistical method was used to predetermine sample size. All cases of infants with Wilms tumours with appropriate material (fresh tissue for organoids or frozen tissue for laser capture microdissection) were investigated. One infant case for which there was no suitable tissue for phylogenetic reconstruction and in which no driver mutation could be found was not described. No other data were excluded from the analyses. The experiments were not randomised. The investigators were not blinded to any features of the dataset. Sex and/or gender were not considered in the study design.

### Organoid culturing and establishment of clonal organoids

Human normal kidney and Wilms tumour organoids were established and maintained using established protocols[15,16]. Following nephrectomy, normal tissue and Wilms tumour tissue was minced into ~1-mm³ pieces and digested in AdDF+++ (advanced DMEM/F12 containing 1× Glutamax, 10 mM HEPES and antibiotics) containing 1 mg ml⁻¹ collagenase (Sigma, C9407) and 10 μM Y-27632 (Abmole) on an orbital shaker for 45 min at 37 °C. Next, the suspensions were washed with AdDF+++ followed by centrifugation at $250 \times g$. The cell pellets were seeded in Basement Membrane Extract (BME) (Bio-Techne) and topped with organoid culture medium (AdDF+++ supplemented with 1.5% B27 supplement (Gibco), 10% R-spondin-conditioned medium, EGF (50 ng ml⁻¹, Peprotech), FGF-10 (100 ng ml⁻¹, Peprotech), N-acetylcysteine (1.25 mM, Sigma), Rho-kinase inhibitor Y-27632 (10 μM, Abmole) and A83-01 (5 μM, Tocris Bioscience).

For maintenance, organoids were dissociated using mechanical dissociation. Following the addition of 5–10 ml AdDF+++ and centrifugation at $250 \times g$, cells were reseeded in BME and topped with organoid culture medium. Medium was changed every 3–4 days, and organoids were passaged every 1–3 weeks.

To obtain clonal normal and tumour organoid cultures, established bulk (i.e. polyclonal) organoid cultures (passage 0/passage 1) were dissociated to single cells using TrypLE Express (Thermo Fisher)

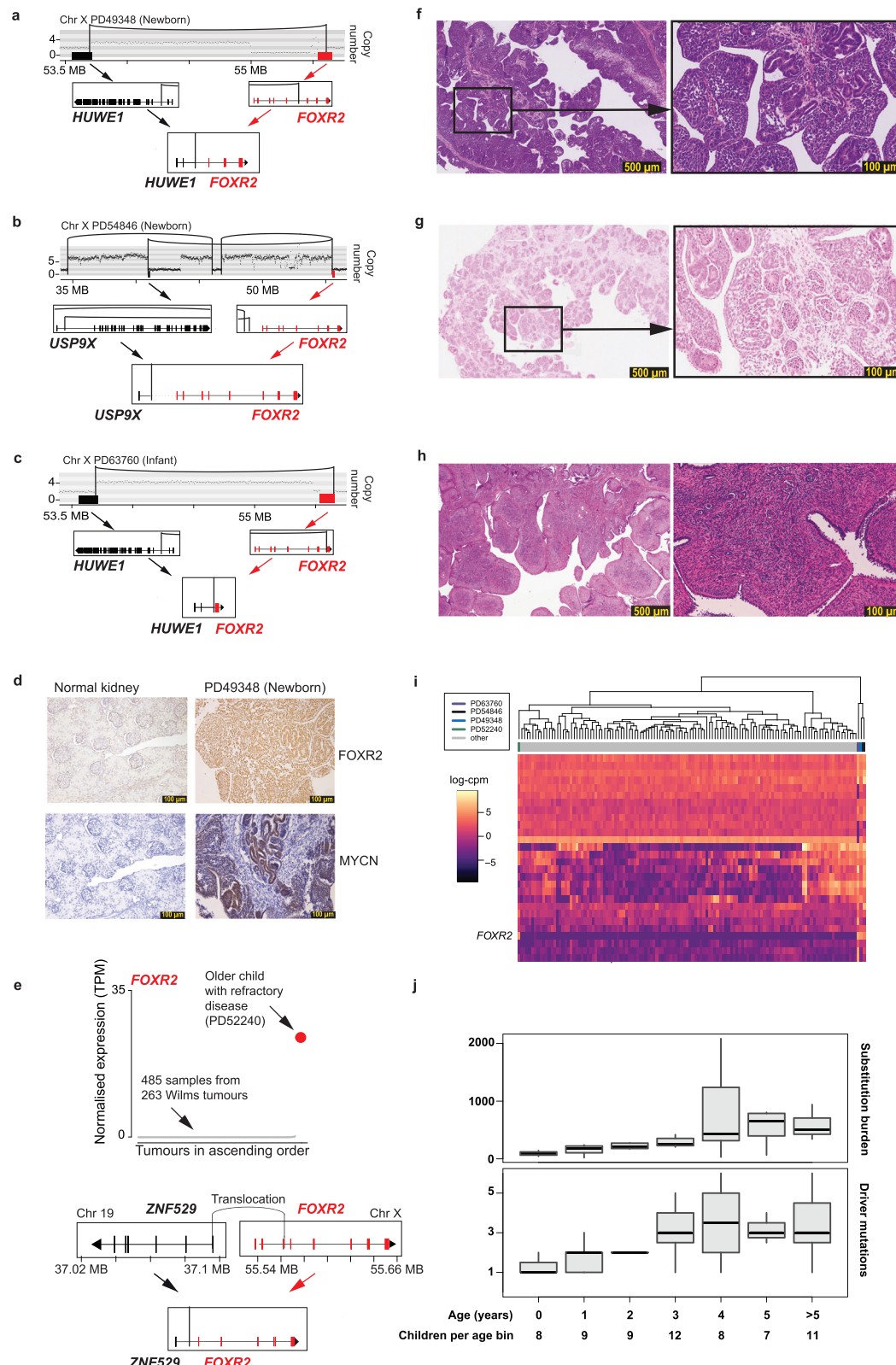

with 10 µM Y-27632 (Tocris Bioscience), washed with and then resuspended in AdDF+++. Single cells were subsequently sorted by flow cytometry and plated in BME and organoid culture medium as described above. After 2–3 weeks, clonal cultures were obtained by manual picking of single-cell-derived individual organoids. Clonal cultures were expanded as described above and used for DNA and RNA extraction (Supplementary Fig. 4).

## Bulk DNA and RNA extraction and sequencing

DNA and RNA were concomitantly extracted from frozen tissue samples or organoid cultures using AllPrep DNA/RNA minikit (Qiagen 80204). For bulk DNA sequencing, short-insert genomic libraries were constructed and 150 base pair paired-end sequencing clusters were generated on the Illumina HiSeq X or Novaseq platforms according to Illumina no-PCR library protocols, and aligned to the human reference

**Fig. 3 | *FOXR2*-mutant Wilms tumours. a−c** For each of three infant Wilms tumours, the structural variant placing *FOXR2* under the control of a different promoter is shown. **d** For an example neonatal Wilms tumour with *FOXR2* rearrangement, FOXR2 and MYCN immunohistochemistry showed strong protein expression compared to normal kidney. Further cases are shown in the Supplementary Note. Immunohistochemistry for each antibody was performed three times with similar results. **e** Whole transcriptomes were generated for 486 samples from a cohort of 264 Wilms tumours and ranked by *FOXR2* expression (TPM, transcripts per million). All cases save one (highlighted in red) did not express *FOXR2* at all. The case with elevated *FOXR2* expression had a complex rearrangement placed *FOXR2* under the control of the promoter of *ZNF529*, shown beneath the RNAseq. **f−h** Haematoxylin and eosin sections showing fibroadenomatoid morphological features of *FOXR2*-rearranged infant Wilms tumours (see Supplementary Note for a further discussion). For each case, the image shown was chosen as representative of fibroadenomatoid architecture based on the examination of a single section of the tumour. **i** Hierarchical clustering of bulk RNAseq for a cohort of Wilms tumours including infant *FOXR2*-rearranged Wilms cases, other infant Wilms tumours, and an older *FOXR2*-rearranged case. Infant *FOXR2*-rearranged cases cluster together and separately from other cases. **j** Comparison of substitution burden and number of annotated driver mutations for a cohort of Wilms tumours by patient age. Only tumours from patients without a known predisposition are included. If there is more than one sample per tumour, the median across all samples is used. For each boxplot the black bar represents the median, the box the interquartile range, and the whiskers extend to the most extreme data point which is no more than the interquartile range from the box. Source data are provided as a Source Data file.

genome using the Burrows-Wheeler Aligner (BWA-MEM)[34]. For RNA sequencing, cDNA libraries were constructed and 75 base pair paired-end sequencing clusters were generated on the Illumina HiSeq 4000. Reads were aligned to the genome using STAR and read counts tabulated using HTSeq[35]. Duplex sequencing libraries were prepared according to the NanoSeq protocol[12]. On-bead fragmentation reaction was carried out with CutSmart buffer (500 mM potassium acetate, 200 mM Tris-acetate, 100 mM magnesium acetate, 1 mg ml−1 BSA, pH 7.9 at 25 °C), HpyCH4V and NFW. A-tailing was carried out with NEBuffer 4, Klenow fragment and dATP/ddBTPs and NFW. Ligation was carried out with xGen Duplex Seq Adapters (IDT, 1080799).

### Laser capture microdissection
Tissue was fixed using PAXgene (Qiagen) and embedded in paraffin. Sections were cut 10 microns thick, and microdissected. Microdissected tissue was lysed using the Arcturus PicoPure Kit (Applied Biosystems) according to the manufacturer's instructions. DNA library preparation proceeded using the low-input protocol[10,18], with fragmentation using NEBNext Ultra II FS Enzyme. Libraries were sequenced on the Illumina Novaseq and aligned to the human reference genome using the Burrows-Wheeler Aligner (BWA-MEM)[34].

### Immunohistochemistry
Tissues and organoids were fixed in 4% paraformaldehyde, dehydrated, and embedded in paraffin. Immunohistochemistry was performed according to standard protocols on 4 μm sections. Sections were subjected to H&E and immunohistochemical staining. The following primary antibodies were used: anti-FOXR2 (Abcam, ab244513, 1:1000) and anti-c-Myc/N-Myc (D3N8F) (Cell Signalling Technology, #13987, 1:200).

### Whole genome sequencing variant calling−substitutions
Substitutions were called using the Cancer Variants through Expectation Maximisation algorithm (CaVEMan, version (version 1.15.1)[36]. For analyses of driver discovery and mutation burden analysis, CaVEMan was run with a matched normal sample. For phylogenetic analyses Caveman was run using an unmatched normal sample (see below).

Post-processing filters included removing common single nucleotide polymorphisms in a panel of 75 unmatched normal samples, passing all soft flags built in to the algorithm[37], and two further filters that remove mis-mapping artefacts: the median alignment score of reads supporting a substitution should be greater than or equal to 140, and fewer than half of these reads should be clipped. For sequences from laser capture microdissected samples, an additional filter that removes artefacts in cruciform DNA structures was applied[18]. For sequences from organoids, only mutations with a variant allele fraction of >0.3 were considered, in order to minimise the contribution of mutations acquired in vitro[17].

To correct for variation in coverage, all samples in the analysis of mutation burden were downsampled to 30X, and CaVEMan was run over the downsampled bam. 30X was chosen as it is the whole genome

coverage that is commonly used in cancer sequencing experiments. Please note that samples from patient PD37585 were the only ones that were not sequenced at >30X (bulk samples were sequenced at 17−20X). Their data has been left uncorrected, and so will represent an underestimate of the 30X mutation burden. Comparison of mutation calls at 30X vs 15X across other samples (Supplementary Fig. 10) suggests that this is not likely to have a very strong effect.

### Whole genome sequencing variant calling−small insertions and deletions (indels)
Indels were called with Pindel[38] (version 3.5.0) and calls passing all in-built soft flags were used.

### Whole genome sequencing variant calling−copy number changes
Copy number changes were detected using the ASCAT algorithm[39].

### Whole genome sequencing variant calling−structural variants
Structural variants were called using the BRASS algorithm (https://github.com/cancerit/BRASS). Reads that paired abnormally were grouped and filtered by read-mapping, removing read-pair clusters that mapped to microbial sequences or where the breakpoint could not be reassembled.

The rearrangement of *FOXR2* was not called by BRASS. This was because of misannotation of the exons of *FOXR2*[27]. The *FOXR2* rearrangement was discovered by examination of RNA-seq data around the copy number change observed in PD49348, which showed that *FOXR2* was overexpressed in the tumour. The *FOXR2* rearrangement was then re-constructed manually by the examination of mis-paired reads on Jbrowse[40]. Reads were later manually assigned to the new exons of *FOXR2* that were described during the preparation of this manuscript[27].

### Duplex sequencing variant calling
Duplex sequencing mutation calling was carried out according to the NanoSeq protocol[12]. We required that each read bundle includes two reads from each of the two original DNA strands, with a consensus base quality score ≥6, and a difference between the best and second best alignment scores of greater than 50. Only two mismatches were allowed per read, and calls within 8 bp of the read end were discarded.

### Cell signal analysis of bulk RNAseq
Bulk RNAseq data was compared to a foetal single cell reference using established methods[41]. This method is based on logistic regression, including an intercept term which represents the 'unexplained signal' (i.e. the signal in the bulk data that cannot be explained well by the single cell reference).

### Construction of phylogenies
Phylogenies were constructed from substitution data using unmatched CaVEMan calls and post-processing as described above. Additional filters were applied for phylogeny construction. In an attempt to

remove germline mutations without removing embryonal mutations, the binomial distribution was used to remove mutations that were clonal in the matched normal sample, and a binomial test was applied to see if the mutation was enriched in the tumour samples relative to the matched normal. Further filters included a binomial test for strand imbalance, and a requirement that each mutation should be called in at least one sample with the following criteria: ≥4 mutant reads, a variant allele fraction ≥0.3, and called on both forward and reverse reads. Variants in regions where the copy number state was not the same for all samples were not included for phylogeny construction. Most Wilms tumours had minimal copy number change, with no mutations removed in any of the infant cases. In PD52209, 4% of mutations were removed, and in PD52230, 1% of mutations were removed. In PD26636, the colorectal cancer, almost all of the genome was at different copy number states in different subsamples. In this case, therefore, mutations were only removed if they occurred in a region with a deletion; this applied to 40% of the genome.

Phylogenies (Supplementary Fig. 5) were either constructed from single-cell-derived organoids (for PD49348, PD37585, and PD26636) or from microbiopsies acquired by laser capture microdissection (for PD54846, PD52209, PD52230, and PD37590). As microbiopsies contained small numbers of cells, were distant from each other, and were largely clonal, each microbiopsy was treated as a clonal unit.

In brief, phylogenies were constructed using maximum parsimony[42], and mutations were assigned to the phylogeny using a maximum likelihood approach[43]. Branch lengths have been scaled in Fig. 2h to account for mutations that were discounted due to copy number changes. To test our confidence in the structure of the tree, the trinucleotide plot of each branch was inspected to look for evidence of artefacts and select mutations from each branch were inspected on Jbrowse.

### Comparison of mutation burden across Wilms tumours

This analysis used previously published data[6,11]. Data from Thatikonda et al. was downloaded from Synapse (ID: syn35289647, https:www.synapse.org/#!Synapse:syn35289647/files/). Data from Coorens et al. was taken from Supplementary Table 4.

### Driver annotation of Wilms tumours

Both somatic and germline variants were annotated. This annotation was agnostic of tumour age. Somatic variants in putative cancer genes were annotated as per tier 1 genes from the Census of Cancer Genes and from a curated list of known Wilms tumour drivers[44–46]. Missense mutations and in frame indels were considered drivers if they occurred in canonical hot spots of oncogenes. As TERT promoter mutations fail Caveman's PASS filter due to being localised in area of simple repeats, these mutations were manually rescued. Truncating mutations were considered drivers if predicted to disrupt the footprint of recessive cancer genes. Focal (<1 MB) homozygous deletions and amplifications (copy number >4 (diploid) or 8 (tetraploid)) in recessive and dominant cancer genes, respectively, were considered drivers. Sub-amplifications in oncogenes were included if accompanied by significantly high RNA expression, as determined by z-score. Rearrangements were considered driver events when they generate a known oncogenic gene fusion or when their breakpoints disrupted the gene footprint of recessive genes. Rearrangements affecting regulatory domains of oncogenes were included if RNA expression was elevated, as per z-score. Rearrangements were validated by manual inspection of both WGS and RNA sequencing data on the genome browser JBrowse to exclude further sequencing artefacts[40].

### Bulk RNA analysis

RNA libraries were sequenced on the Illumina HiSeq 4000 platform. Reads were aligned using STAR, and mapped to GRCh37, and read counts of genes were obtained using HTSeq[47,48]. Data was processed in R using EdgeR, normalised using the TMM method and converted to log-CPM values[49]. Differential gene expression analysis was performed using Limma and Glimma, requiring a log fold change significantly >1[50,51]. For differential gene expression analysis, we compared primary tumour samples to a published cohort of 107 children with Wilms tumours[52], keeping only samples with a tumour cell content above 60%, as estimated by Battenberg[53]. The Euclidean distance between samples was calculated based on the differentially expressed genes, and only genes with a gene symbol were included, before unsupervised clustering.

### Reporting summary

Further information on research design is available in the Nature Portfolio Reporting Summary linked to this article.

## Data availability

The DNA sequencing data generated in this study have been deposited in the European Genome-Phenome Archive (EGA) under accession code EGAD00001009812. Raw RNA data generated in this study have been deposited in the European Genome-Phenome Archive (EGA) under accession code EGAD00001009813 and EGAS00001005244. This is a permanent repository. The raw data are available under restricted access due to data privacy laws. Access may be granted following an application to the Data Access Committee (datasharing@sanger.ac.uk). A response is usually provided within 3 weeks. Once a request is approved, raw data may be downloaded and used for the duration of time agreed with the Data Access Committee. Publicly available processed data used in this study are available at are available on Synapse (https://www.synapse.org/) under the id 'syn35289647' for[6] and within the supplementary information file provided with the manuscript for[13]. No restricted access agreement is needed to use these data. Data used for preparing figures are provided in the Source Data file. Source data are provided with this paper.

## Code availability

The R scripts used to run the bespoke analyses detailed in this study can be found here: https://github.com/HLee-Six/infant_Wilms (https://doi.org/10.5281/zenodo.15045557)[54].

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

## Acknowledgements

This study was supported by the Wellcome Trust (core grant 206194, Senior Research Fellowship 223135/Z/21/Z), The Little Princess Trust (CCLGA 2019 27A), a European Research Council (ERC) starting grant (grant number 850571), and the Dutch Cancer Society (KWF)/Alpe d'HuZes Bas Mulder Award (grant number 10218). The SIOP WT 2001 trial was managed by the University of Birmingham Cancer Research UK Clinical Trials Unit (CRCTU) and funded by Cancer Research UK (grants C1010/A2889; C1188/A8687) and supported by the UK National Cancer Research Network and the Children's Cancer and Leukaemia Group (CCLG). We thank Hua Pan and Steve Baker at CRCTU for their help accessing the data. The IMPORT study is funded by the Children's Cancer and Leukaemia Group/Little Princess Trust (grant refs: CCLGA 2019 10 and CCLGA 2019 27) and received past funding from Children's Cancer and Leukaemia Group/Bethany's Wish (grant ref: CCLGA 2017 02), EU-FP7 grant refs: 261474 (ENCCA) and 270089 (P-medicine), Great

Ormond Street Children's Charity (grant ref: W1090) and Cancer Research UK (grant ref: C1188/A17297) and benefits from the infrastructural support of the UK National Cancer Research Network and the CCLG. HLS was funded by the NIHR on an Academic Foundation Programme and Academic Clinical Fellowship and is the recipient of a Junior Research Fellowship at Trinity College, Cambridge. THHC is the recipient of an EMBO long-term fellowship (ALTF 172-2022). AW is funded by the Wenner-Gren Foundations. KPJ is funded in part by the National Institute for Health Research GOSH University College London Biomedical Research Centre, Great Ormond Street Hospital (GOSH). We are grateful to Aidan Maartens for his helpful comments on our manuscript. Lastly, we thank the children and their families for participating in our research.

## Author contributions

S.B. and J.D. designed the experiment, with the assistance of H.L.S. H.L.S. and M.D. carried out laser capture microdissection. H.L.S. and T.D.T. analysed the data, with the assistance of T.H.H.C., N.D.A., A.W. and Y.W. Y.T., S.D. and S.D.H. cultured organoids. M.M.v.d.H.E., R.A.S., A.La., A.Le., J.W., C.P., M.G., T.C., M.R.S. and K.P.J. provided expertise and contributed to patient recruitment and sample identification and management. G.M., G.V., M.J.O., R.R.d.K. and J.C.H. provided pathological expertise and selected cases. H.L.S. and S.B. wrote the manuscript. S.B. and J.D. co-directed the study.

## Competing interests

The authors declare no competing interests.
