## [Transparent Peer Review file · Nature Communications]

High resolution clonal architecture of hypomutated Wilms tumours

Corresponding Author: Professor Sam Behjati

Version 0:

Reviewer comments:

Reviewer #1

(Remarks to the Author)

Thank you for addressing all of my comments in detail. The manuscript has considerably improved.

(Remarks on code availability)

Reviewer #2

(Remarks to the Author)

I thank and congratulate the authors for their thoughtful responses to our prior review. The paper is very well written and easy to follow. The associations with the FOXR2 structural variant add to the paper. The authors have also nicely shown, by adding additional sequencing of matched normal kidney, that the differences in mutation rates that they observe are specific to the infant setting. I have no other concerns and look forward to publication of this manuscript.

(Remarks on code availability)

Reviewer #3

(Remarks to the Author)

In this paper, Lee-Six et al utilizes duplex genomic sequencing to identify increased number of mutations compared to conventional methods; then found clonality differences between low and high mutated tumors using spatial laser capture microdissection. Through this work, the authors now highlight a specific subtype of Wilms tumor that has a FOXR2 mutation.

The authors have expanded on their initial study that was reviewed. From a technical and methodological angle, the authors have demonstrated the utility of the nanoseq assay in pediatric tumors that are characteristically considered to have low TMB, expanding these findings from their adult experience. This raises a valuable consideration for the application of this technique in other pediatric tumors with classically low TMB, including rhabdoid tumors et al. The authors also add additional information that characterizes a particular infantile WT with FOXR2.

Based on the additional information provided, I feel the authors have satisfactorily responded to the reviewer comments. It would be highly recommended that the authors consider renaming their manuscript though, as the statement of childhood renal cancers is still too broad. If the authors added others such as translocation RCC or MRT, this title is fair. Otherwise, it should be limited to WT.

(Remarks on code availability)
